# Analysis of Self-Reported Walking for Transit in a Sprawling Urban Metropolitan Area in the Western U.S.

**Courtney Coughenour [1],\*** 🆔**, Hanns de la Fuente-Mella [2] and Alexander Paz [3]**

[1] School of Community Health Sciences, University of Nevada Las Vegas 4505 S. Maryland Pkwy,
Box 3064 Las Vegas, NV 89154, USA

[2] Escuela de Comercio, Facultad de Ciencias Económicas y Administrativas, Pontificia Universidad Católica
de Valparaíso 2340031, Chile; hanns.delafuente@pucv.cl

[3] School of Civil Engineering and Built Environment, Queensland University of Technology, 2 George Street,
Brisbane QLD 4000, Australia; alexander.paz@qut.edu.au

\* Correspondence: courtney.coughenour@unlv.edu

**Abstract:** Walkability is associated with increased levels of physical activity and improved health and sustainability. The sprawling design of many metropolitan areas of the western U.S., such as Las Vegas, influences their walkability. The purpose of this study was to consider sprawl characteristics along with well-known correlates of walkability to determine what factors influence self-reported minutes of active transportation. Residents from four neighborhoods in the Las Vegas Metropolitan Area, targeted for their high and low walkability scores, were surveyed for their perceptions of street-connectivity, residential-density, land-use mix, and retail–floor-area ratio and sprawl characteristics including distance between crosswalks, single-entry-communities, high-speed streets, shade, and access to transit. A Poisson model provided the best estimates for minutes of active transportation and explained 11.28% of the variance. The model that included sprawl characteristics resulted in a better estimate of minutes of active transportation compared to the model without them. The results indicate that increasing walkability in urban areas such as Las Vegas requires an explicit consideration of its sprawl characteristics. Not taking such design characteristics into account may result in the underestimation of the influence of sprawl on active transportation and may result in a missed opportunity to increase walking. Understanding the correlates of walkability at the local level is important in successfully promoting walking as a means to increase active transportation and improve community health and sustainability.

**Keywords:** active living; physical activity; walkability; active transportation; public health; utilitarian activity

---

## 1. Introduction

The link between physical activity and health is well established, and participating in regular physical activity improves overall health, controls weight, reduces the risk of chronic disease, and improves psychological well-being. However, in 2017, only about 50% of American adults participated in enough aerobic physical activity to meet the 150 min per week recommended by the Centers for Disease Control and Prevention [1]. Physical inactivity is a major public health concern, as it is correlated with reduced overall health and wellbeing, increased rates of chronic disease and shortened life expectancy [2].

Current efforts to increase rates of physical activity in the United States have focused on walking, including a Surgeon General's Call to Action to Promote Walking and Walkable Communities in

2015 [3]. There are two basic categories of walking: recreational walking and utilitarian walking. Recreational walking is done purposefully to obtain exercise; it is a conscious decision and requires a high level of commitment. Utilitarian walking most often refers to walking for transportation or active transport. Active transport is defined as physical activity that occurs incidentally while accomplishing another purpose; for example, walking to work or completing errands [4]. Promoting active transport may prove more effective and successful as a means to reducing physical inactivity because it does not require an intense level of commitment, which often causes an individual to abandon such behaviors [5,6]. This is an important distinction, as the amenities and built environment to promote recreational walking and active transport are likely to differ significantly. Walking for active transport may be a key component of attaining the recommended amount of physical activity.

Poor air quality is associated with negative health outcomes, and there may therefore be concern that some benefits associated with active transport may be offset by the exposure to pollution. Although air pollution does have negative health implications, other recent studies have found that the benefits of physical activity outweigh the potential negative implications. Kubesch and colleagues (2018) reported that "the beneficial effects of physical activities on incident and recurrent myocardial infarctions (MI) are independent of the exposure to NO2, and are not reduced in those living in areas with high residential NO2 levels. Thus, the long-term benefits of physical activity in preventing the development of MI in healthy, middle-aged participants, and possibly as effective disease control in patients with prior MI, can outweigh the risks associated with enhanced residential exposure to traffic-related air pollution during physical activity" [7]. Similarly, Andersen and colleagues (2015) stated that "overall, the long-term benefits of physical activity in terms of reduced mortality outweigh the risk associated with enhanced exposure to air pollution during physical activity"[8]. Fortunately, the use of green technologies such as electric vehicles and solar power is on the rise, and air pollution may decrease in the near future as a result. Relatedly, the more active transport, such as walking, that takes place, fewer emissions are released.

Neighborhood walkability is a measure of how conducive the built environment is to walking. High measures of neighborhood walkability have been associated with increased minutes of physical activity [9–12], more minutes of active transport each week [9,13,14], and a lower body mass index (BMI) [15–17]. In addition to health benefits, active transport is a more sustainable form of transportation; it has been estimated that switching from vehicular travel to active modes of transit would result in improved air quality [18,19]. Thus, removing barriers to walking and promoting neighborhood design to enhance walkability is one approach to reducing physical inactivity, increasing rates of active transport, and improving overall sustainability.

*Sprawl*

Sprawl is an urban design characteristic that undermines walkability. It is "characterized by low densities, spatially segregated land uses and a street network with low connectivity" [20]. Ewing et al. found that increased sprawl was associated with fewer minutes walked [21,22] as well as higher traffic and pedestrian fatalities [23,24]. Additionally, sprawl is a less sustainable form of development, as it "requires more infrastructures since it takes more roads, pipes, cables and wires to service these low-density areas compared to more compact developments with the same number of households" [25]. Design features that are associated with sprawl are relevant to walkability; however, these characteristics are either not captured by many of the standard walkability measures, or they may be measured as the same in theory, but are not similar in actual design. Specifically, land-use mix is associated with more active transport in traditional neighborhoods because individuals have convenient access to various amenities. However, due to the predominance of single-use zoning, a sprawling neighborhood that is measured as high land-use mix would likely consist of a single family residential development abutted to a block wall with a strip-mall type development of various land uses on the other side. This is illustrated in Figure 1, which contrasts high land-use in a more traditional (older) metropolitan area to a sprawling metropolitan area.

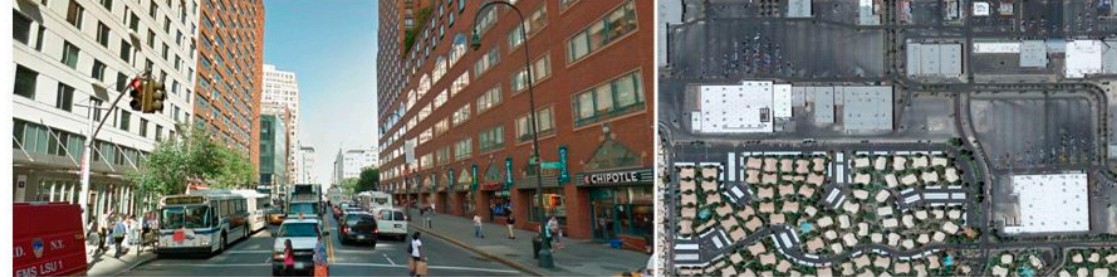

East 14th Street, New York, NY
*Images from Google maps, 2013

South Jones Boulevard, Las Vegas, NV

**Figure 1.** Mixed land-use may look different in sprawling cities when compared to older, non-auto-centric cities.

Typical street networks in sprawling communities contain multiple high-speed arterial streets. High-speed streets are not only unpleasant for pedestrians, but also result in decreased safety [26]. Additionally, this design may result in long distances between crosswalks, which often exist at intersections of arterial streets that can be one mile apart or greater. This makes active transport inconvenient and reduces pedestrian safety. Further, nested between these high-speed arterial streets are numerous "common interest" housing developments. Many of these housing developments are gated or single-entry communities that have high (six feet or higher) cement block privacy walls surrounding the entire sub-division [27]. The single or limited-access points both in and out of the developments make active transport inconvenient by significantly reducing route options and increasing the trip distance (see Figure 2). One study examined the role of single entry communities on street connectivity and concluded that measuring street connectivity without taking them into account resulted in an over-estimation [28]. Another factor that plays a role in active transport is convenient access to public transit. Research posits that those with public transit options near their homes are more likely to use it [29], and those who take public transit utilize more active transport and attain more minutes of physical activity [30,31]. The aforementioned characteristics of sprawl negatively influence public transit use; as Fulton et al. note, sprawling neighborhoods "are built at densities that make it difficult to provide public transit alternatives" [32]. Measuring walkability without taking sprawl characteristics into account is likely to undermine the importance of the built environment on walking.

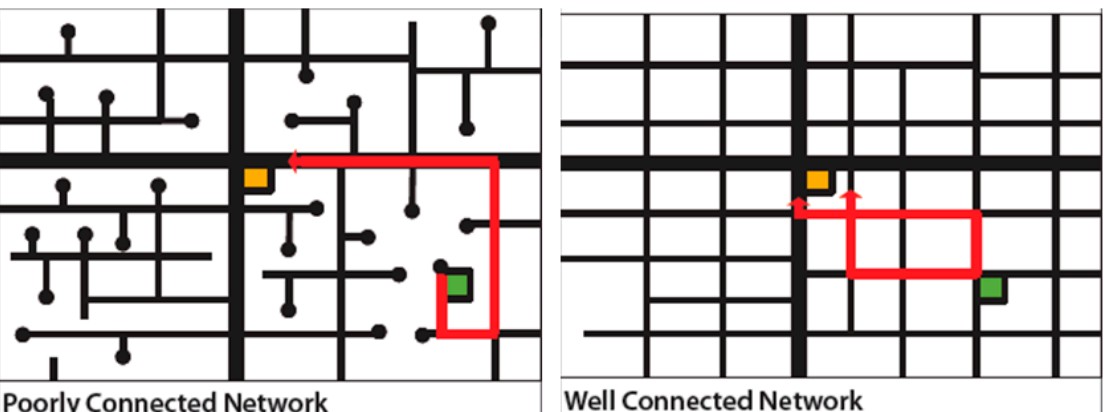

**Poorly Connected Network**

**Well Connected Network**

**Figure 2.** A well-connected street network with more intersections reduces trip distance and makes active travel more convenient. Source: [33].

The Las Vegas Metropolitan Area (LVMA) is one such sprawling city worth studying, as it possesses many of the sprawl characteristics significant to active transport and walkability.

Additionally, understanding region-specific design characteristics is critical in order to properly focus prevention and intervention efforts, as well as allocate funding and resources towards such efforts. One design characteristic of particular importance in LVMA is the use of shade trees to minimize the effects of the extreme desert heat. In Las Vegas, the daily high temperature exceeds an average of over 100 °F for three months out of the year; having too few trees for shade would make active transport unpleasant. The objective of this study was to determine if incorporating sprawl design characteristics of specific importance to LVMA (high-speed streets, distance between crosswalks, single-entry communities, shade, and access to transit), in addition to well-known correlates of walkability in the home neighborhood, would result in a better estimation of self-reported minutes of active transport per week in this region. To meet this objective, survey data from targeted neighborhoods with a known walkability index score were collected. Two Poisson regression models were used—one with only the standard index variables and one with the standard index variables plus the LVMA design characteristics—in order to determine which model was a better fit for self-reported minutes of active transport per week. To the best of our knowledge, this type of analysis has not been conducted previously in LVMA.

## 2. Materials and Methods

### 2.1. Setting

LVMA is a newer metropolitan area in the southwestern United States with an urban form that is characteristic of sprawl. LVMA's rapid population growth coincided with the automobile age; consequently, the Las Vegas metropolitan area is auto-centric, caters to the automobile, and lacks an older 'core'.

### 2.2. Neighborhood Selection

Seven neighborhoods across LVMA, defined as a census block group, were chosen based on geographic location. These neighborhoods were limited to those with 450 to 700 households and those with a median household income between $38,521 and $101,582, according to 2010–2014 estimates of the American Community Survey (ACS) [34]. The size of the census block group was limited to this number of households to ensure that the methodology used for door-to-door sampling was feasible. Income brackets were chosen because they represented the third and fourth income quintiles for the 2011 U.S. Census [35]. Neighborhoods were limited to moderate income levels, as the intention was to determine which built-environment characteristics might influence walking. Studies have shown that low-income individuals are more likely to be 'captive walkers'—walking not out of choice, but out of necessity—because they do not have access to a private vehicle. Such individuals are more likely to walk regardless of the design of the built environment [36,37]. Thus, limiting the study to moderate-income neighborhoods was an attempt to limit the number of captive walkers and increase the number of respondents whose decision to walk is influenced by the built environment.

### 2.3. Walkability Index

The index developed by Frank et al. (2010) was used to calculate neighborhood walkability for each of the seven census block groups [38]. This index was chosen because it is used widely in the literature [9,10,12,13,15,17,38] and has high construct validity [38]. This methodology consisted of utilizing geographic information systems (GIS) to measure four components of the built environment: net residential density, retail–floor-area ratio, intersection density, and land-use mix, defined in Table 1. Data were retrieved from parcel-based land-use data and street center-line data supplied by the Clark County Assessor's Office. The number of residential units were calculated by summing the number of single-family homes and units in multifamily homes; residential density was calculated by dividing the number of residential units by the land area devoted to residential development per neighborhood. The retail–floor-area ratio was calculated by dividing the retail building area foot-print by the land area

devoted to retail. Land-use mix was calculated using the entropy index methodology developed by Cerin and colleagues; see Equation (1) [39]. The "entropy index ranges from 0 to 1, with 0 representing homogeneity (all land uses are of a single type), and Equation 1 representing maximal heterogeneity (the land use categories are evenly represented) [39]".

$$ - \frac{\sum_k p_k \ln(p_k)}{\ln(N)} \tag{1} $$

where $k$ denotes the land use category, $p$ is the proportion of the land area within a neighborhood allocated to a specific land use, and $N$ is the number of land use categories [39].

The number of intersections were calculated using a street network analysis in GIS to create point shapefiles at each intersection; intersection density was calculated by dividing the number of intersections by the land area in each neighborhood. The land area for each measure was calculated using the "calculate geometry" function in GIS. The calculated values for the four built environment components were normalized using the Z-score formula outlined in Equation (2), and then summed.

$$ \frac{x - \mu}{\sigma} \tag{2} $$

**Table 1.** Definitions of each component of the walkability index as defined by Frank et al., 2010 [38].

| Component | Definition |
|---|---|
| Residential density | Number of residential units divided by land area in acres devoted to residential use. |
| Intersection density | Number of true intersections (3 or more segments) divided by the land area of the block group in acres. A higher ratio indicates greater connectivity. |
| Land-use mix | Diversity of land-use types in a block group. Land-use types include retail, residential, entertainment (parks, recreation facilities, theatres, restaurants), office, and institutional (schools, religious institutions, libraries/museums, community organizations, government facilities). Values were normalized between 0 and 1, with 0 being single use and 1 indicating a completely even distribution of land area. |
| Retail–floor-area ratio | Retail building area foot-print divided by retail land area foot-print. The higher the ratio is, the more indicative it is of pedestrian friendliness. |

The Z-score value of intersection density was weighted by a factor of '2' because street connectivity—a measure of a more direct path—has a strong influence on walking behaviors. The weighting scheme was further confirmed by Frank and colleagues (2005) who described the index methodology in detail [38]. The walkability index was the sum of Z-scores of the four built environment components (Equation (3)) [38]. The raw scores and Z-scores for each walkability score component are shown in Table 2.

$$ 2 \cdot Z - \text{intersection density} + Z - \text{net residential density} + Z - \text{retail floor area ratio} + Z - \text{land} - \text{use mix [38]} \tag{3} $$

**Table 2.** Four-component walkability index scores for seven census block groups (neighborhoods) in the Las Vegas Metropolitan Area.

| Neighborhood | Residential Density | | Intersection Density | | Retail–floor Area Ratio | | Entropy Score (Land-Use Mix) | | Walkability score | |
|---|---|---|---|---|---|---|---|---|---|---|
| | Raw Score | Z score | Raw Score | 2x Z score | Raw score | Z score | Raw Score | Z score | | |
| NW | 9.33 | −0.1 | 0.13 | −3.02 | 0 | −1.37 | 0 | −1.14 | −5.64 | Low |
| S | 7.79 | −1.05 | 0.26 | 0.55 | 0 | −1.37 | 0 | −1.14 | −3.02 | Low |
| 1 | 9.91 | 0.25 | 0.20 | −1.10 | 0.311 | 0.54 | 0.699 | 0.86 | 0.56 | Medium |
| 2 | 7.38 | −1.3 | 0.29 | 1.37 | 0.43 | 1.28 | 0.147 | −0.72 | 0.63 | Medium |
| 3 | 9.45 | −0.03 | 0.22 | −0.55 | 0.28 | 0.37 | 0.812 | 1.19 | 0.98 | Medium |
| SE | 10.38 | 0.54 | 0.22 | −0.55 | 0.27 | 0.29 | 0.692 | 0.84 | 1.13 | High |
| E | 12.23 | 1.688 | 0.36 | 3.30 | 0.27 | 0.27 | 0.439 | 0.12 | 5.36 | High |

Neighborhoods were arranged in ascending order—the least to the most walkable—based on their walkability index scores. Residents of the two most walkable neighborhoods—those with the highest index scores—and two least walkable neighborhoods—those with the lowest index scores—were targeted for surveying (see Figure 3). The neighborhoods with the highest and lowest walkability scores were selected in an attempt to capture neighborhoods that differed by vetted correlates of walkability.

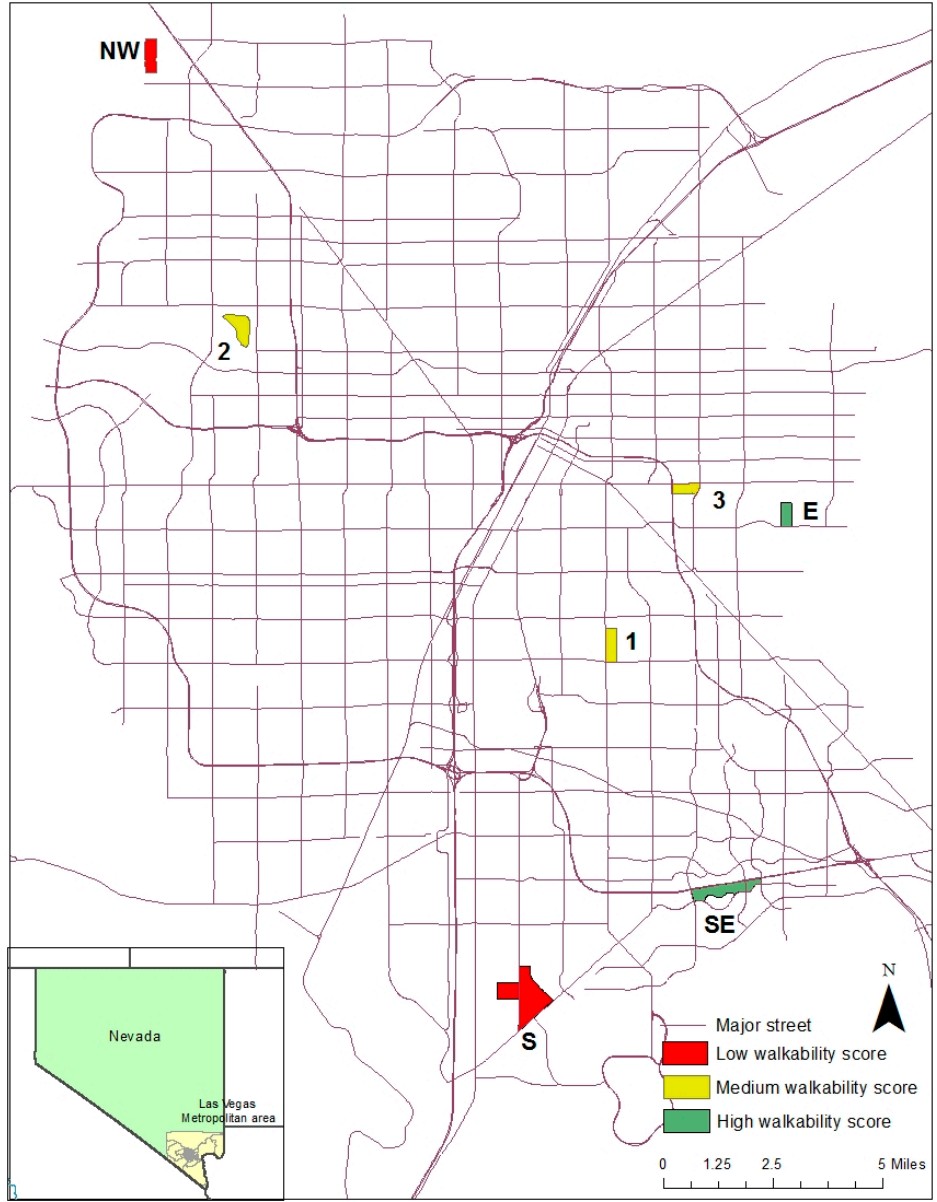

**Figure 3.** Map of Las Vegas Metropolitan Area (LVMA) neighborhoods in which the walkability index score was calculated, and the four neighborhoods which were targeted for surveying (2 low and 2 high walkability scores).

## 2.4. Sprawl Design Characteristics

Google Maps and satellite imaging available in ArcGIS was used to determine objective measures for each of the five sprawl design characteristics of specific importance to LVMA for each of the four surveyed neighborhoods. High-speed streets were defined as the total km of road network with speeds greater than 35 mph; a long distance between crosswalks were defined by the total number of instances in which a crosswalk was not present for greater than 0.25 miles of arterial roadway; single-entry

communities were defined as the percentage of residences located inside of a single-entry development; shade was defined as the percentage of the neighborhood that had shade cover; and access to transit was defined as the number of transit stops per square km.

### 2.5. Measures

A survey was created as a tool to measure the association between perceptions of home neighborhood walkability and weekly minutes of active transport. The survey inquired about sociodemographic characteristics (age, race, gender, and education), length of residence in the neighborhood, car ownership, residence in a gated community, and whether the respondent had an impairment or health problem that limited their ability to walk. Respondents who answered 'yes' to having an impairment that limited their ability to walk were removed from the analysis.

To determine how respondents perceived their home neighborhood design, questions were created regarding the four walkability index characteristics—residential density, intersection density, land-use mix and retail–floor-area ratio—and five design characteristics of sprawl in LVMA—distance between crosswalks, single-entry communities, shade, high-speed streets, and access to transit—as a barrier to walking for active transport in their home neighborhood. The full list of questions aimed at determining perceptions of neighborhood attributes as barriers to walking for transport can be found in Appendix A. The questions were asked using a reverse-coded six-point Likert scale, ranging from 'strongly disagree' to 'strongly agree'.

To determine if perceptions of home neighborhood design were associated with minutes of physical activity, a modified version of the International Physical Activity Questionnaire (IPAQ) was used to measure weekly minutes of active transport. Respondents reported the number of hours and minutes that they typically walked for transportation each day and the total number of days per week that they typically walked for transportation. Weekly minutes of active transport were determined by multiplying minutes walked per day by the number of days they typically walked (see Appendix B). The IPAQ has been previously validated. In a review of the literature, Lee et al. (2011) reported that most studies validated the use of the IPAQ against accelerometer data for walking with acceptable correlation effect sizes between 0.4 and 0.79 [40]. This revised questionnaire was developed for adults aged 18 to 65 to measure four domains: (1) transportation, (2) work, (3) leisure, and (4) household and gardening. Questions regarding work and household gardening were removed for this study, as the actual work and household environments were not assessed. Leisure time physical activity was not included in analysis, and only minutes of active transport were analyzed.

The survey was distributed to each residence within the four neighborhoods in the fall of 2012 by hanging a printed version on each door. A cover letter in both English and Spanish requested that at least one resident per household between the ages of 18 and 65 years participate in the survey by 1) completing the printed version and returning it in a supplied, postage paid, business reply envelope, or 2) visiting a website to complete an electronic version of the survey hosted by Qualtrics. The paper version of this questionnaire was distributed in English; in addition, there was an option for a Spanish version if the participant completed the survey electronically. Participants were offered compensation for their time by means of entry into a prize-draw to win prizes and gift cards. As a reminder, a cardboard door hanger was distributed approximately two weeks after the initial survey to each residence, asking them to return their completed paper survey if they had not already done so, or complete the electronic version of the survey. This protocol was deemed exempt by the University of Nevada, Las Vegas Office of Research Integrity.

### 2.6. Analysis

Self-reported weekly minutes of active transport and perceptions of the four walkability index characteristics and five design characteristics of sprawl were calculated from survey responses. Two Poisson regression models were used for statistical analysis—one with only the standard index variables and one with the standard index variables plus the sprawl design characteristics—in order

to determine which model was a better fit for minutes of active transport per week. In this case, the dependent variable $Y$ (minutes of active transport per week) was limited to non-negative integers; as a result, a model for count data, Poisson regression, was used. In some cases, where the dependent variable is strictly positive, it is possible to log-transform the data and use a linear model. However, for this study, the dependent variable had a high percentage of zero values (36.4%), and 2% of the cases took the highest values. Therefore, the Poisson regression model, which ensures positive values for all independent variables and for all parameter values, was the best fit.

In order to improve the quality of the estimates, multiple models for count data were tested including Poisson, negative binomial, exponential, zero-inflated and normal/nonlinear least squares. The Poisson regression model resulted in the best pseudo R-squared, Akaike, Schwarz, and Hannan–Quinn criteria.

## 3. Results

### 3.1. Descriptive Analysis

From 2227 residences within the four targeted neighborhoods, 147 surveys were completed and returned for a 6.5% response rate. Most of the surveys were returned by mail (72%). Three surveys were removed from the analysis due to reporting 'yes' to having a disability that limited their ability to walk, resulting in a total sample size of 144.

In a demographic breakdown of survey responses (Table 3), the mean age of respondents was 41.5 years and the mean length of residence in the neighborhood was 5.8 years. Most respondents reported that they owned a vehicle (92%) and 49% reported that they lived in a single-entry community. Our sample had slightly more respondents who lived in a single-entry community than the general population of the Western United States, as the American Housing Survey [AHS] (2009) reported that 39% of respondents in the Western U.S. lived in a community that was secured by walls or fences [41]. Unfortunately, the AHS estimates are unavailable at the geographical scale of LVMA alone. About 96% of homes in LVMA report that they have a vehicle available to them, similar to 92% of respondents. Of note, survey respondents were more educated than the general LVMA population, with 54.2% of the respondents having a four-year college degree or greater compared to 22.2% for LVMA residents [42]. This was not surprising, however, given that only moderate-income neighborhoods were targeted and that income is highly correlated with education [43]. Respondents were asked if they perceived each design characteristic as a barrier to walking in their neighborhood. Results by neighborhood are listed in Table 3. The mean number of self-reported minutes of active transport per week was 169.9 min. The relationship between resident perceptions of each design characteristic and objectively measured characteristics for each of the four neighborhoods was investigated using the Spearman's rho correlation coefficient. The majority of objectively measured variables were correlated with respondent perceptions. The results are presented in Table 4.

**Table 3.** Census median household income, respondent demographics, and percent of respondents from the Las Vegas Metropolitan Area who agreed or strongly agreed that the design characteristics prevent them from walking for active transport (AT).

| | NW (*n* = 35) | S (*n* = 31) | SE (*n* = 58) | E (*n* = 20) | Total (*n* = 144) |
|---|---|---|---|---|---|
| Median household income * | $67,609 | $78,810 | $84,545 | $41,500 | |
| Race | | | | | |
| White | 60.0% | 58.1% | 79.3% | 50.0% | 66.0% |
| Non-white | 40.0% | 41.9% | 20.7% | 50.0% | 34.0% |
| Gender | | | | | |
| Female | 61.8% | 64.5% | 75.9% | 70.0% | 69.2% |
| Male | 38.2% | 35.5% | 24.1% | 30.0% | 30.8% |
| Education | | | | | |
| Less than college | 51.4% | 54.8% | 27.6% | 75. 0% | 45.8% |
| Four-year degree or greater | 48.6% | 25.8% | 72.4% | 25.0% | 54.2% |
| Age | | | | | |
| 18–29 years | 17.1% | 29.0% | 9.0% | 25.0% | 17.7% |
| 30–39 years | 25.7 % | 35.4% | 16.3% | 25.0% | 24.1% |
| 40–49 years | 42.9% | 22.6% | 32.7% | 35.0% | 33.3% |
| 50–59 years | 14.2% | 6.5% | 32.7% | 15.0% | 19.9% |
| 60–64 years | 0% | 6.5% | 9.0% | 0% | 5.0% |
| Lack of shade prevents walking for AT | 44.1% | 53.1% | 25.0% | 50.0% | 39.9% |
| Single-entry communities prevents walking for AT | 23.5% | 9.4% | 8.9% | 9.1% | 12.6% |
| Distance between crosswalks prevent walking for AT | 8.8% | 40.6% | 1.8% | 45.5% | 18.9% |
| High-speed streets prevent walking for AT | 26.5% | 43.8% | 14% | 22.7% | 25.0% |
| Large parking lots prevent walking for AT | 33.3% | 18.8% | 8.8% | 31.8% | 21.0% |
| Poor land-use mix prevent walking for AT | 38.2% | 81.3% | 12.5% | 40.9% | 39.2% |
| Poor street connectivity prevents walking for AT | 29.4% | 62.5% | 14.3% | 40.9% | 32.2% |
| Poor residential density prevents walking for AT | 5.9% | 28.1% | 1.8% | 22.7% | 11.9% |
| Convenient access to transit result in greater amounts of walking | 23.5% | 46.9% | 16.1% | 36.4% | 26.6% |

NW = northwest, S = south, SE = southeast, E = east; * US Census Bureau's American Community Survey 2010–5 yr estimates.

**Table 4.** Spearman rho correlations between perceived neighborhood design characteristics and objectively measured neighborhood design characteristics *n* = 144.

| | Perceived Measures (Reverse Coded) |
|---|---|
| Objectively measured retail–floor-area ratio | −0.521 ** |
| Objectively measured land-use mix | −0.561 ** |
| Objectively measured intersection density | 0.231 ** |
| Objectively measured residential density | −0.085 |
| Objectively measured number of single entry communities | 0.122 |
| Objectively measured percent shade cover | −0.299 ** |
| Objectively measured number of high speed streets (>35 mph) | −0.217 * |
| Objectively measured long distance between crosswalks (arterial roads with >0.25 miles to cross) | −0.246 ** |
| Objectively measured number of transit stops | 0.003 |

* $p \leq 0.05$, ** $p \leq 0.01$.

### 3.2. Poisson Regression Model

For the model that contained the four components of the standard index and the five sprawl design characteristics, the results included individual significance levels and joint levels for model variables at 99%, with the exception of the variable age, which was not significant. The variability of the endogenous variable was explained in 11.28% of the sample by the variability of the exogenous

variables. Additionally, the model presented adequate information criteria including Akaike, Schwarz, and Hannan–Quinn (Table 5).

**Table 5.** Results for the Poisson models containing perceptions of 1) the four components of the standard walkability index and five sprawl design characteristics estimating minutes of active transport per week; 2) only the four components of the standard walkability index estimating minutes of active transport per week (*n* = 137).

| (1) Variables | Coefficient | Std. Error | Z-Statistic | *p*-Value |
|---|---|---|---|---|
| Education (0 = less than four-year degree; 1 = four-year degree or greater) | −0.358 | 0.014 | −26.154 | <0.001 |
| Gender (0 = female; 1 = male) | 0.213 | 0.015 | 14.173 | <0.001 |
| Race (0 = white; 1 = non-white) | 0.373 | 0.015 | 24.872 | <0.001 |
| Age | 0.000 | 0.001 | 0.620 | 0.5351 |
| Street connectivity | −0.252 | 0.007 | −34.886 | <0.001 |
| Distance between crosswalks | −0.021 | 0.007 | −3.102 | 0.002 |
| Land-use mix | 0.085 | 0.004 | 21.108 | <0.001 |
| Residential density | 0.135 | 0.008 | 17.196 | <0.001 |
| Retail–floor-area ratio | −0.086 | 0.006 | −13.286 | <0.001 |
| Tree shade | 0.237 | 0.004 | 55.781 | <0.001 |
| Single-entry community | −0.080 | 0.006 | −13.309 | <0.001 |
| High-speed streets | −0.033 | 0.007 | −4.838 | <0.001 |
| Convenient access to transit * | 0.164 | 0.005 | 36.890 | <0.001 |

1) *p*-value = 0.002; pseudo R-squared: 0.113; Akaike: 301.851; Schwarz: 302.149; Hannan–Quinn: 301.972

| (2) Variables | Coefficient | Std. Error | z-Statistic | *p*-Value |
|---|---|---|---|---|
| Education (0 = less than four-year degree; 1 = four-year degree or greater) | −0.357 | 0.013 | −27.026 | <0.001 |
| Gender (0 = female; 1 = male) | 0.296 | 0.014 | 20.532 | <0.001 |
| Race (0 = white; 1 = non-white) | 0.562 | 0.014 | 40.736 | <0.001 |
| Age | 0.002 | 0.001 | 2.270 | 0.023 |
| Street connectivity | −0.179 | 0.007 | −26.754 | <0.001 |
| Land-use mix | 0.141 | 0.004 | 37.335 | <0.001 |
| Residential density | 0.052 | 0.007 | 7.061 | <0.001 |
| Retail–floor-area ratio | −0.005 | 0.006 | −0.764 | 0.445 |

2) *p*-value = 0.023; Pseudo R-squared: 0.054; Akaike: 336.137; Schwarz: 336.329; Hannan–Quinn: 336.215. * Reverse coded.

The signs for the coefficients in Table 5 are as expected. For example, a negative sign for the variable "distance between crosswalks" indicates that the more the perceived distance between crosswalks is, the less likely people are to report walking for active transport. Similarly, a negative sign for "high-speed streets" is associated with a decreased likelihood of reporting active transport. In contrast, a positive sign for "convenient access to transit" indicates that the more perceived access to transit there is, the more likely people are to report walking for active transport.

Those who perceived their neighborhoods as being less connected (fewer intersections), having long distances between crosswalks, with larger parking lots to accommodate vehicles (i.e., not pedestrian-friendly), high-speed streets, and a high number of single-entry communities reported fewer minutes of active transport. Perceptions of inadequate shade from the sun, low land-use mix, and low residential density were associated with increased minutes of active transport. Non-white individuals and those with lower educational attainment reported more minutes of active transport. Including the sprawl design characteristics of specific importance to LVMA into the regression model created a better estimator of minutes of active transport than the model that only included the four standard characteristics.

For the model that contained the four components of the standard walkability index, results included the individual significance levels and the joint levels for model variables at 95%, with the

exception of the variable retail–floor-area ratio (i.e., pedestrian friendliness), which was not significant. The variability of the endogenous variable was explained only in 5.37% of the sample by the variability of the exogenous variables. Additionally, the model presented adequate information criteria (Akaike, Schwarz, Hannan–Quinn) (Table 4).

Various specifications were tested including Poisson, negative binomial, exponential, zero-inflated and normal/nonlinear least squares. Overall, the Poisson model provided a better goodness-of-fit and revealed significant variables (data not shown).

## 4. Discussion

The study results showed that incorporating sprawl design characteristics of specific importance to LVMA in addition to well-known correlates of walkability resulted in a better estimation of self-reported minutes of active transport per week. This finding highlights the importance of recognizing correlates of walking and walkability at the local geographic level. Failing to take certain environmental features into consideration may result in an underestimation of the influence that the built environment has on active transport. Additionally, a clear understanding of the role of such characteristics is necessary for transportation engineers, urban planners, and public health professionals who aim to increase rates of physical activity through active transport.

Survey results indicated that residents perceived neighborhood design characteristics as a barrier to walking. Specifically, a lack of shade was reported as a barrier in most neighborhoods, as nearly 40% reported that this prevents them from walking for active transport. Poor land-use mix and street connectivity were both reported as a barrier by a large percentage of respondents. There were some notable differences in the percentage of agreement between neighborhoods, illustrating that there are some neighborhood designs in LVMA that are less conducive to walking than others.

Similar to previous findings, those who perceived their neighborhoods to be more connected reported more minutes of active transport [17,44]. This is likely because increased connectivity results in shorter trip distances as distance and convenience are strongly associated with transportation choices [45,46]. Likewise, perceptions of a high number of single-entry communities and long distances between crosswalks were associated with fewer minutes of active transport, both of which would increase distance and reduce convenience.

Larger parking lots are a proxy measure for an environment that caters to the automobile, resulting in a non-pedestrian friendly environment. Respondents who perceived large parking lots reported fewer minutes of active transport. Correspondingly, high-speed streets result in reduced pedestrian friendliness, and respondents who perceived more high-speed streets in their neighborhood reported fewer minutes of active transport. Using public transportation often is considered a means of active transport because the individual walks to and from the transit line during each trip. Using public transportation has been associated with significantly more minutes of physical activity [47].

Surprisingly, those who perceived inadequate amounts of shade reported more minutes of active transport. This is an unexpected finding given the desert climate and the high percentage of respondents who reported lack of shade as a barrier to walking. This may be a reflection of self-selection bias; those individuals who were already interested in walking and not deterred from the lack of shade may have elected to complete the survey. Low land-use mix and low residential density were associated with more active transport. One potential explanation for this variation from previous findings may be because a high land-use mix in LVMA, and other sprawling metropolitan areas, may be measured as the same but is likely to differ in actual design from more traditional (older) neighborhoods. As pointed out previously in Figure 1, neighborhoods with a high land-use mix in LVMA are likely to consist of single-family residential development abutted to a block wall with a strip-mall type development on the other side of that wall. This design, in practice, does not necessarily increase the convenience or perceived safety related to access to services.

Similarly, residential density may differ as well. Residential lots in LVMA are small in comparison to non-western states [48] for a number of reasons, including the arid climate, which naturally lends

itself to small lot sizes, and the growth constriction of the surrounding mountains and federally owned land [49]. Consequently, the population density of LVMA is dense, with about 4500 people per square mile [50], yet it still has single-use zoning and a predominance of single-family developments [51]. As a result, the existing infrastructure prohibits capitalization of this density to an environment that encourages active transport.

The study findings confirm that sprawl has a negative effect on active transport in our sample, which is likely to translate into negative health and safety outcomes as well. Efforts to manage sprawl in a sustainable way are critical. Given the pervasiveness of this design standard, efforts must be multifaceted, interdisciplinary, and unafraid of repurposing existing infrastructure. For example, a critical first step is convenient and efficient alternative transportation, yet this needs to be done in collaboration with housing and transportation stakeholders to ensure that housing affordability is maintained and that existing roadway infrastructure can accommodate all users, while giving priority to the more efficient mode.

*Limitations of This Study*

Several important limitations need to be considered. The sample size in this study was relatively small, with a total of 144 respondents. The small sample size may have masked significant results due to increases in Type II error, and may result in a possible non-response bias. In other words, those who completed the survey may be different than those who did not complete the survey [52]. Self-selection bias may have confounded results; it may be that those who were already interested in or enthusiastic about walking were more apt to complete the survey than those who are not concerned with walking [53]. The authors can only speculate on the reasons for the small sample, but the time compensation of prizes and gift cards may not have been incentive enough, or perhaps respondents felt that the process to complete the survey was not convenient enough. Knocking on doors and asking for in-person completion may have yielded a higher response rate, although that would have required more time and resources. However, even with the small sample, it is estimated that the 93 completed and valid surveys provide a 50% probability that the sample would present the characteristics of the larger population [54]. Considering this criterion, we exceeded the minimum number of samples with 144 responses. Although the literature supports the relationship between the built environment and physical activity [9–15], it may be the case that some study participants choose to walk, or not walk, regardless of the built environment. This study relied on self-reported minutes of active transport, although the most commonly-used measure of physical activity, self-reported data, has been found to differ from objectively measured data [55–57]. The model explained a small percentage of the variance, which is common in built-environment and physical-activity literature. For example, Frank and colleagues present a rational explanation for this, stating that "a wide variety of demographic, biological, psychological, behavioral, social, and environmental variables are correlated with physical activity, so it is not expected that any single variable or set of variables will explain large amounts of variance. Even when many variables are included in multivariate models, it is uncommon to explain >30% of the variance in physical activity" [12]. They posit that "untested environmental variables will explain additional variance" [12].

## 5. Conclusions

In conclusion, the current study adds to the literature by examining the role that design characteristics of sprawl have on minutes of active transport. This study adds value in that we found that incorporating design characteristics of sprawl with established correlates of walkability resulted in a better estimation of self-reported minutes of active transport per week. This illustrates that LVMA, and likely similar sprawling southwest metropolitan areas, are unique in a way that may require different approaches and strategies aimed at increasing walkability. Not taking such design characteristics into account may result in underestimating the influence of sprawl on active transport and may result in a missed opportunity to increase rates. There is a role for the design professions and

policy makers to take actions aimed at urban development in such a way that improves walkability and takes the pedestrian into account. Smart growth policies and complete streets policies are examples of best practices that take all road users into account and can serve as a guide for new development as well as retrofitting efforts. Local efforts should begin to advocate for the adoption of such policies. When implementing such design standards, it is important that local stakeholders take into account some of the unique issues faced by the pervasiveness of previous design standards that favored urban sprawl. Future research that confirms the current findings and expands the examination of neighborhood walkability and the effects of sprawl on health outcomes and sustainability is warranted.

Increasing walkability is vital to pedestrian safety, environmental friendliness, sustainability, and community health. For public health efforts to effectively increase active transport by means of improved walkability, it is important first to determine which built environment characteristics are associated with walking at the local level. Multifaceted and collaborative efforts between transportation engineers, public health professionals, urban planners, and policy makers to develop a model for walkability in sprawling communities and embrace sustainable efforts that manage sprawl are timely and significant.

**Author Contributions:** Conceptualization, C.C. and A.P.; methodology, C.C.; formal analysis, H.d.l.F.-M.; investigation, C.C.; resources, C.C.; data curation, C.C.; writing—original draft preparation, C.C.; writing—review and editing, C.C., H.d.l.F.-M., A.P.; visualization, C.C., H.d.l.F.-M.; project administration, C.C.

**Funding:** This research received no external funding.

**Acknowledgments:** The publication fees for this article were supported by the UNLV University Libraries Open Article Fund.

**Conflicts of Interest:** The authors declare no conflict of interest.

## Appendix A

**Table A1.** Survey questions measuring perceptions of standard walkability characteristics and design characteristics of sprawl as barriers to walking for transport.

- There are not enough trees that provide shade cover in my neighborhood which keep me from walking from place to place during the summer months.
- There are multiple single entry communities in my neighborhood (either gated or un-gated communities which have only one entrance in and out of the complex) which keep me from walking from place to place.
- The long distance between crosswalks on major streets in my neighborhood keep me from walking from place to place.
- The presence of high speed, multiple lane streets in my neighborhood keep me from walking from place to place.
- The retail areas and stores in my neighborhood have large parking lots and are not pedestrian friendly which keep me from walking from place to place.
- There is not a good mix of stores and facilities (ie: entertainment/restaurants, retail, offices, schools) in my neighborhood which keep me from walking from place to place.
- There are not many alternative routes for getting from place to place in my neighborhood (I have to go the same way every time because the streets are not well connected) which keep me from walking from place to place.
- There are not many housing units (houses, townhomes, and apartments/condos) and/or the housing units are too spread apart in my neighborhood which keep me from walking from place to place.
- If it was convenient to access to public transit (ie: bus stops) in my neighborhood, I would walk more often.

Note: * place to place was defined as "any walking you may do secondary to other activities, such as going to work, the store, or running errands".

## Appendix B

A survey question from the International Physical Activity Questionnaire (IPAQ) was used to measure weekly minutes spent walking for transportation.

During the last 7 days, on how many days did you walk for at least 10 min at a time to go from place to place?
\_\_\_\_\_\_\_ days per week or \_\_\_\_none
How much time did you usually spend on ONE of those days walking from place to place?
\_\_\_\_\_ hours per day \_\_\_\_\_\_ minutes per day

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
