# Peer review of "Analysis of Self-Reported Walking for Transit in a Sprawling Urban Metropolitan Area in the Western U.S."

_sustainability, doi:10.3390/su11030852_

Round 1

Reviewer 1 Report

This paper analyses the self-reported walking for transit in four neighborhoods of Las Vegas Metropolitan Area. I have some major concerns:

Response      rate is very low and so is % of variance explained by the model. How are the      results significant with such a low response rate and low % of variance      explained? Are results really meaningful and tell a complete picture?      Unexpected findings on those who perceived inadequate amounts of shades      but reported more minutes of active transport may be a reflection of      insignificance of the results beside self-selection.

I lived      in a desert environment for couple of years. Evenings are usually      pleasant. It is possible that most of the people walk in the evenings and      it may have nothing to do with the built environment elements. Authors did      not acknowledge it or mentioned it anywhere.

Introduction:      It is good, provides overview of walkability and sprawl. I was wondering      if there was no previous study done on walkability, sprawl or built      environment characteristics in LVMA?

Material      and Methods: Researchers totally rely on walkability index developed by      Frank et al. which is fine but failed to provide walkability index for      four neighborhoods. How did they calculate it? More details must be      provided on the actual calculation of index with values of built      environment characteristics. How was GIS used in the calculations?

“Z-score      value of intersection density was weighted by a factor 2 because street      connectivity ... had a strong influence on walking behavior”. There is no      argument or justification to back it up. It may be true in some areas but      not for entire western US.

There      should also be a paragraph or two in Introduction or Discussion comparing      walkability index (developed by Frank et al) and other walkability indices      out there. There are many in the literature why did authors prefer to use Frank’s?

Figure 1      is not of high quality. Authors mentioned the use of ArcGIS. Much better      map can be created with ArcGIS. Add scale. Mention what yellow polygons      are. Add detailed map of each neighborhood with built environment elements      used to calculate walkability.

In survey      (appendix), there was no question on number of minutes people walk. Where      this information came from?

Walkability      certainly has positive impact on health but recent studies show that air      pollution can offset the health benefits of walking. Authors should      acknowledge it and if available provide some information on LVMA air      quality.

Minor Comments:

Line 29:      not sure what is meant by “increasing rates”.

Line45:      Should add a reference here.

Table 1:      What is the role of floor area in Land-use mix? It seems last sentence is      for Retail Floor Area.

Line 167:      ArcGIS Imaging? What is imaging? There is an extension of ArcGIS as Image      analyst. Did authors use it?

Line      295-96, same sentence as on 267-68. Remove it.

Author Response

Comment:

“This paper analyses the self-reported walking for transit in four neighborhoods of Las Vegas Metropolitan Area. I have some major concerns:

Response rate is very low and so is % of variance explained by the model. How are the results significant with such a low response rate and low % of variance explained? Are results really meaningful and tell a complete picture? Unexpected findings on those who perceived inadequate amounts of shades but reported more minutes of active transport may be a reflection of insignificance of the results beside self-selection.

Response:

These types of results are actually common in built environ and physical activity research. For example, a study by Frank et al (reference below) calculated a model that explained 10% of the variance. The explanation they gave states that “A wide variety of demographic, biological, psychological, behavioral, social, and environmental variables are correlated with physical activity, so it is not expected that any single variable or set of variables will explain large amounts of variance. Even when many variables are included in multivariate models, it is uncommon to explain >30% of the variance in physical activity”.  The following paragraph was added to the limitations section.

“The model explained a small percentage of the variance, which has been seen before in the built environment and physical activity literature. For example, Frank and colleagues present a rational explanation for this, stating that “a wide variety of demographic, biological, psychological, behavioral, social, and environmental variables are correlated with physical activity, so it is not expected that any single variable or set of variables will explain large amounts of variance. Even when many variables are included in multivariate models, it is uncommon to explain >30% of the variance in physical activity [10].”

Frank, L.D.; Schmid, T.L.; Sallis, J.F.; Chapman, J.; Saelens, B.E. Linking Objectively Measured Physical Activity with Objectively Measured Urban Form: Findings from SMARTRAQ. Am. J. Prev. Med. 2005, 28, 117-125.

Comment:

“I lived  in a desert environment for couple of years. Evenings are usually pleasant. It is possible that most of the people walk in the evenings and  it may have nothing to do with the built environment elements. Authors did not acknowledge it or mentioned it anywhere.”

Response:

The following sentence noting this has been added in the limitations section.

“Although literature supports the relationship between the built environment and physical activity [7-12], it may be the case that some study participants choose to walk, or not walk, regardless of the built environment. This study relied on self-reported minutes of active transport, and, although the most commonly used measure of physical activity, self-reported data has been found to differ from objectively measured data [52-54].”

Comment:

Introduction: It is good, provides overview of walkability and sprawl. I was wondering  if there was no previous study done on walkability, sprawl or built environment characteristics in LVMA?

Response:

Correct – we have added the following sentence at the end of the introduction:

“To the best of our knowledge, this type of analysis has not been conducted previously in LVMA.”

Comment:

Material  and Methods: Researchers totally rely on walkability index developed by  Frank et al. which is fine but failed to provide walkability index for  four neighborhoods. How did they calculate it? More details must be  provided on the actual calculation of index with values of built  environment characteristics. How was GIS used in the calculations?

Response:

A table has been added that has the values for each of the walkability score components for the 4 neighborhoods. More details on the calculation of each component has been added to section 2.2. It reads:

“Data were retrieved from parcel-based land-use data and street center-line data supplied by the Clark County Assessor’s Office. The number of residential units were calculated by summing the number of single family and units in multifamily homes; residential density was calculated by dividing the number of residential units by the land area devoted to residential development per neighborhood. Retail floor area ratio was calculated by dividing the retail building area foot print by the land area devoted to retail. Land-use mix was calculated using the entropy index methodology developed by Cerin and colleagues, see equation 1 [37]. The “entropy index ranges from 0 to 1, with 0 representing homogeneity (all land uses are of a single type), and 1 representing maximal heterogeneity (the land use categories are evenly represented) [37].” The number of intersections were calculated using a street network analysis in GIS to create point shapefiles at each intersection; intersection density was calculated by dividing the number of intersections by the land area in each neighborhood. The land area for each measure was calculated using the “calculate geometry” function in GIS. The calculated values for the four built environment components were normalized using the Z-score formula outlined in equation 2, and then summed.”

Comment:

 “Z-score value of intersection density was weighted by a factor 2 because street connectivity had a strong influence on walking behavior”. There is no argument or justification to back it up. It may be true in some areas but not for entire western US.

Response:

The weighting is a component of the same methodology used throughout the paper, developed by Frank et. al. (2005). A notation of their confirmation of this weighting scheme has been added. It now reads:

“The Z-score value of intersection density was weighted by a factor of ‘2’ because street connectivity – a measure of a more direct path – has a strong influence on walking behaviors. The weighting scheme was further confirmed by Frank et. al. (2005) who described the index methodology in detail [36].

Comment:

There should also be a paragraph or two in Introduction or Discussion comparing walkability index (developed by Frank et al) and other walkability indices out there. There are many in the literature why did authors prefer to use Frank’s?

Response:

We selected Frank’s because it is the most used in the literature. A statement about this was added in Section 2.3. Now it reads:

“2.3 Walkability index

The index developed by Frank et al. (2010) was used to calculate neighborhood walkability for each of the seven census block groups [38]. This index was chosen because it is used widely in the literature [9,10,12,13,15,17,38] and has high construct validity [38]. This methodology consisted of utilizing geographic information systems (GIS) to measure four components of the built environment: net residential density, retail floor area ratio, intersection density, and land-use mix, defined in Table 1.”

Comment:

Figure 1 is not of high quality. Authors mentioned the use of ArcGIS. Much better  map can be created with ArcGIS. Add scale. Mention what yellow polygons  are. Add detailed map of each neighborhood with built environment elements  used to calculate walkability.

Response:

The map figure was improved. The authors felt that the level of scale for the map was not useful to represent each index measure, but feel that by providing the scores for each index measure by neighborhood (Table 2 – which is newly added), the same context is provided.

Comment:

In survey   (appendix), there was no question on number of minutes people walk. Where this information came from?

Response:

This survey question has been added to the appendix

Comment:

Walkability certainly has positive impact on health but recent studies show that air pollution can offset the health benefits of walking. Authors should  acknowledge it and if available provide some information on LVMA air quality.

Response:

The following material was added to the paper:

“Poor air quality is associated with negative health outcomes, and thus, there may be concern that some benefits associated with active transport may be offset by the exposure to pollution. Although air pollution does have negative health implications, other recent studies have found that the benefits of physical activity outweigh the potential negative implications. Kubesch and colleagues (2018) reported that “The beneficial effects of physical activities on incident and recurrent myocardial infarctions (MI) are independent of the exposure to NO2, and are not reduced in those living in areas with high residential NO2 levels. Thus, the long-term benefits of physical activity in preventing the development of MI in healthy, middle-aged participants, and possibly as effective disease control in patients with prior MI, can outweigh the risks associated with enhanced residential exposure to traffic-related air pollution during physical activity”[7]. Similarly, Andersen and colleagues (2015) stated that “Overall, the long-term benefits of physical activity in terms of reduced mortality outweigh the risk associated with enhanced exposure to air pollution during physical activity”[8]. Fortunately, use of green technologies such as electric vehicles and solar power is on the rise, and air pollution may decrease in the near future as a result. Relatedly, the more active transportation such as walking that takes place, the fewer emissions released.”

 Comment:

Minor Comments:

Line 29:      not sure what is meant by “increasing rates”.

Response:

Has been changed to say ‘increase walking”

Line45:      Should add a reference here.

Response:

The reference to the US Surgeon General’s call to action has been added.

Table 1:      What is the role of floor area in Land-use mix? It seems last sentence is  for Retail Floor Area.

Response:

This is the terminology used by Frank et. al. (2005), though replacing the term “floor” with “land” enhances clarity, and thus, we have done that.

Line 167:      ArcGIS Imaging? What is imaging? There is an extension of ArcGIS as Image      analyst. Did authors use it?

Response:

We used satellite imagery from ArcGIS, no specific extension. It has been rephrased to read:

“Google maps and satellite imaging available in ArcGIS was used to determine objective measures for each of the five sprawl design characteristics of specific importance to LVMA for each of the four surveyed neighborhoods.”

Line      295-96, same sentence as on 267-68. Remove it.

Response:

We have removed the duplicative sentence.

Reviewer 2 Report

A very interesting article with outstanding results, I have only three small comments:

In the introduction section, Figure 1, why the authors did not use two images from Las Vegas, since this is the region under study.

With the construction of walkability, it would be interesting to include a map on the spatial variation of the index at the LVMA scale.

Finally, could the authors add some explanations for the low response rate?

Author Response

 Comment:

A very interesting article with outstanding results, I have only three small comments:

In the introduction section, Figure 1, why the authors did not use two images from Las Vegas, since this is the region under study.

Response:

Thank you for this inquiry. The authors were trying to contrast differing city design by showing Las Vegas next to a more traditional city design with mixed land use such as New York. We have stated this directly in the paper and the sentence now reads “This is illustrated in Figure 1, which contrasts high land-use in a more traditional (older) metropolitan area to a sprawling metropolitan area.”

 Comment:

With the construction of walkability, it would be interesting to include a map on the spatial variation of the index at the LVMA scale.

Response:

The authors included table 2 which now shows the index scores for all neighborhoods measured, and figure 3 illustrates the corresponding neighborhoods.

 Comment:

Finally, could the authors add some explanations for the low response rate?

Response:

We did add a sentence speculating on the reasons for the low response. It now reads “The authors can only speculate on reasons for the small sample, but the time compensation of prizes and gift cards may not have been incentive enough, or perhaps it is that respondents felt that the process to complete the survey was not convenient enough. Knocking on doors and asking for in person completion may have yielding a higher response rate, though would have required more time and resources. However, even with the small sample, it is estimated that 93 completed and valid surveys provide a 50% probability that the sample would present the characteristics of the larger population [57]. Considering this criteria, we exceed the minimum number of samples with 144 responses.

Reviewer 3 Report

A very interesting study of great social significance. The impact of physical activity on health and well-being is obvious and not subject to discussion. However, we do not always realize what factors can affect the reduction of physical activity in society. If, thanks to this type of research, we learn that one of such factors may be the urban configuration of the area, it seems that the role of city authorities seems to be taking actions aimed at planning development in such a way that they are more friendly to people.

The authors rightly noticed the limitations of the study. The most important of them is the number of respondents. It is worth continuing this research taking into account the activities aimed at obtaining opinions from a larger number of respondents.

Author Response

Comment:

A very interesting study of great social significance. The impact of physical activity on health and well-being is obvious and not subject to discussion. However, we do not always realize what factors can affect the reduction of physical activity in society. If, thanks to this type of research, we learn that one of such factors may be the urban configuration of the area, it seems that the role of city authorities seems to be taking actions aimed at planning development in such a way that they are more friendly to people.

The authors rightly noticed the limitations of the study. The most important of them is the number of respondents. It is worth continuing this research taking into account the activities aimed at obtaining opinions from a larger number of respondents.

Response:

Thank you. We agree with the reviewer that further studies are required to continue getting insights and increasing our understanding about how the built environment affects health.

We did add a sentence speculating on the reasons for the low response. It now reads “The authors can only speculate on reasons for the small sample, but the time compensation of prizes and gift cards may not have been incentive enough, or perhaps it is that respondents felt that the process to complete the survey was not convenient enough. Knocking on doors and asking for in person completion may have yielding a higher response rate, though would have required more time and resources. However, even with the small sample, it is estimated that 93 completed and valid surveys provide a 50% probability that the sample would present the characteristics of the larger population [57]. Considering this criteria, we exceed the minimum number of samples with 144 responses.

Round 2

Reviewer 1 Report

Authors have addressed all concerns. No further comments.